# Determinants of Communication Failure in Intubated Critically Ill Patients: A Qualitative Phenomenological Study from the Perspective of Critical Care Nurses

**DOI:** 10.3390/healthcare11192645

**Published:** 2023-09-28

**Authors:** Catalina Perelló-Campaner, Antonio González-Trujillo, Carme Alorda-Terrassa, Maite González-Gascúe, Josep Antoni Pérez-Castelló, José Miguel Morales-Asencio, Jesús Molina-Mula

**Affiliations:** 1Emergency Care Service 061, 07011 Palma, Spain; 2SATSE CIDEFIB, c/Antoni Marques, 4. Bjs izqda, 07003 Palma, Spain; 3Emergency Hospital Care Service, Hospital de Manacor, 07500 Manacor, Spain; 4Nursing and Physiotherapy Department, University of Balearic Islands, 07122 Palma, Spainjesus.molina@uib.es (J.M.-M.); 5Intensive Care Unit, University Hospital Son Espases, 07120 Palma, Spain; 6Psychology Department, University of Balearic Islands, 07122 Palma, Spain; 7Universidad de Málaga, Faculty of Health Sciences, Department of Nursing, Instituto de Investigación Biomédica de Málaga (IBIMA-Bionand), 29016 Málaga, Spain

**Keywords:** communication aids for disabled, communication barriers, critical care, intensive care unit, intubation, intratracheal, nonverbal communication, nurse–patient relations, respiration, artificial, tracheostomy, nurses

## Abstract

Aim: To explore what factors determine communication with awake intubated critically ill patients from the point of view of critical care nursing professionals. Background: Impaired communication frequently affects mechanically ventilated patients with artificial airways in the intensive care unit. Consequences of communication breaches comprise emotional and ethical aspects as well as clinical safety, affecting both patients and their conversation partners. Identification of determining factors in communication with awake intubated patients is needed to design effective action strategies. Design: A qualitative phenomenological approach was used. Methods: Semi-structured interviews were used as the data collection method. A total of 11 participants from three intensive care units of three Majorcan public hospitals, selected by purposive sampling, were interviewed. Findings: Three major themes regarding the communication determinants of the awake intubated critically ill patients were identified from the interviewees’ statements: factors related to the patient (physical and cognitive functionality to communicate, their relational and communicative style and their personal circumstances), to the context (family presence, ICU characteristics, workload, availability/adequacy of communication aids, features of the messages and communication situations) and, finally, those related to the professionals themselves (professional experience and person-centredness). Conclusions: The present study reveals determinants that influence communication with the awake intubated patient, as there are attitudes and professional beliefs. Relevance to clinical practice: The discovery of relations between different kinds of determinants (of patient, context or professionals) provides a multi-factor perspective on the communicative problem which should be considered in the design of new approaches to improve communicative effectiveness. This study is reported according to the COREQ checklist.

## 1. Introduction

Impaired communication frequently affects mechanically ventilated patients with an artificial airway in the intensive care unit (ICU). Since the 1980s, several studies identified difficult oral communication as one of the main stressors of these patients, with little progress so far [1,2,3,4,5,6]. The inability to express oneself and to be understood increases when, in addition to the incapacity of speech, the patient is limited in other ways of communication such as writing, gestures or mouthing [7].

Nursing professionals are the main conversation partners for the critically ill patient with impaired communication. Nevertheless, interactions usually are short, started and controlled by nurses, driven by tasks or procedures and reduced to orders, closed-ended questions or provision of information [7,8,9,10].

Consequences of the communication breach encompass emotional reactions, clinical safety issues and ethical aspects. Emotional effects include a wide range of feelings such as frustration, despair, fear and terror, anxiety, loss of control, insecurity, loneliness, rage, helplessness and even apathy [1,3,4,6,11,12,13,14,15,16,17,18]. Similar reactions have been described by patients’ next of kin and health care professionals [19,20,21,22]. These negative emotions lead to resignation or avoidance of communication attempts by patients and/or their conversation partners [13,14,15,20,21,23].

Furthermore, communication difficulties may compromise patients’ clinical safety. Bartlett et al. [24] evidenced that patients with this condition showed a higher risk of avoidable multiple adverse effects during hospital admission. Likewise, misreading of the critically ill patient’s communication attempts may lead to failures to identify their immediate needs and to inappropriate and/or unnecessary treatments [25].

Finally, communication failure involves the patient’s inability to exercise their right to autonomy and to state their care or treatment preferences [13,17].

The communication difficulty of awake intubated patients in the ICU is determined by many factors. Borsig and Steinacker [26] classified the communication barriers in five categories: psychological or physical, social, chemical, environmental and organic/therapeutic. Nearly a decade after, Albarran [27] added the professionals’ lack of training in alternative communication methods and strategies as an additional barrier.

In the last decade, very few studies have been specifically focused to reveal these communication determinants [21,28,29,30,31]. In a review, Dithole et al. [32] identified five factors which impact on communication with these patients: the patient’s level of consciousness and/or physical condition, the kinds of nurse–patient interactions, availability and use of assisted communication methods, the professionals’ skills and perceptions, as well as the physical environmental features of the ICU. However, they did not explore their interrelations and possible patterns, which explain the communication problems with intubated patients. Furthermore, the methodological heterogeneity of the included studies hindered the integration of the findings. Thus, more research is needed to develop effective strategies to improve communication with those patients and reduce the negative effects of this issue.

The aim of this study was to investigate what factors determine communication of awake intubated critically ill patients in the ICU from the perspective of the nursing professionals who care for them, due to their key role in the majority of interactions with these patients. A more comprehensive understanding of the factors influencing communication with these individuals (initiation/termination/maintenance of the communicative act, duration, effectiveness, content, etc.) can inform the design of new and more suitable strategies to address this phenomenon.

## 2. Methods

### 2.1. Design

A qualitative study with a phenomenological approach was conducted, as the fundamental nature of the meanings attributed by humans is only accessible through internal subjectivity [33]. Moreover, phenomenology considers the context in which people interact, forming the basis for constructing the sense and meaning of discourses [34].

Communication with awake, intubated critically ill patients is an everyday element of nursing practice in ICUs. Therefore, exploring the lived experience from the perspective of nursing professionals not only contributes to a description of the foundational aspects of the communicative phenomenon, but also facilitates a deeper exploration of the significance of (in)communication for these patients and for their conversational partners. In this regard, interpretative phenomenology provided a suitable philosophical foundation for the present research [35].

### 2.2. Participants Recruitment

Sample collection was carried out through purposive sampling of maximum variation [36] in the three ICUs involved in the study. An initial number of 8 participants (nurses and nursing assistants) was set, with a possibility to extend it depending on data saturation. The inclusion criteria were the following: (1) minimum professional experience of 6 months in the present ICU and (2) accept taking part in the study and signing the informed consent. In the progressive selection of participants, convenience and sufficiency criteria were implemented to grant appropriateness and a diversity of views, thus broadening the obtained discourse on the studied phenomenon [36].

Contact with participants was made by members of the research team working in the units under study. Possible candidates who met the inclusion criteria were identified, and verbal information about the project was provided to them. If they agreed to participate, their contact information was collected, and an interview appointment was scheduled through a phone call.

Finally, the initial sample size was expanded to 11, at which point data saturation was reached in the analysis.

### 2.3. Settings

The study was carried out in three public hospitals on Majorca island (Spain), with each of their adult ICUs having a minimum of 6 beds: a third-level hospital (HSE), the Balearic Islands reference hospital (822 beds, 32 of them for critical care) and two general hospitals (HSLL and HFM, 422 and 238 beds, 18 and 6 ICU beds, respectively) [37]. The units involved different organisational features (type of patients, staff allocation, family visit regimes); thus, maximum diversity of the contextual factors was assured. The multidisciplinary team did not include speech therapists.

### 2.4. Data Collection

The data were collected through semi-structured interviews by the principal researcher (PC), a nurse (RN, MSc) trained in qualitative methods and with extensive experience in critically ill patient care. Previous elaboration of the initial interview script was based on a review of the existing literature [38]. Two pilot interviews, not incorporated in the analysis, were conducted prior to data collection to test whether the questions were clear, to detect the need for modifications and to allow the interviewer to become familiar with the script [39] (for further details about this process, see Appendix A).

Once the participant accepted to take part in the study, the main researcher contacted him by phone and made an appointment for the interview. At the beginning of the interview, the researcher introduced herself, offered written and verbal information on the study and the researchers’ interests on the topic, reminded the participant that participation was voluntary and revocable at any moment, as well as that the interview would be audio recorded. At that stage, the participant signed the informed consent and the investigator carried on collecting socio-demographic data. The researcher and participant had not met before. Every participant could provide any additional information by email if considered.

The interviews were conducted in a quiet place chosen by the interviewees. Interviews lasted an average of 33 min. Strategies to maximise data collection were used: flexibility, active listening with minimum intervention, sensitivity to the interviewee and the discourse and clarification of concepts, among others [39,40,41]. The interviewer kept a listening attitude and the participant led the interview development. Each participant was interviewed once, without the presence of any other person than the investigator.

Field notes were taken with non-verbal cues, characteristics of the interview process, or any data to complement the verbal discourse [41].

### 2.5. Ethical Considerations

Approval for the study was granted by the Ethics Committee of Clinical Investigation at the Balearic Islands (IB 1700/11 PI). We also obtained permits from the centres involved.

No participant revoked consent neither before nor after the interview.

### 2.6. Data Analysis

Data analysis was initiated after transcription of the first interviews and continued concurrently with the incorporation of new participants during the study [42] until data saturation was achieved. We defined data saturation, following Morse’s framework, as the construction of rich data within the research process, considering its scope (breadth and depth of data) and repetition. This approach aims to provide a comprehensive, cohesive and in-depth understanding of the studied phenomenon [43]. The analysis was carried out following Taylor and Bodgan’s principles in three different stages: discovery of the data, elaboration of codes and categories and relativisation of the findings [40].

Discovery of the data involved transcription of the interviews and field notes, review and comparison of the transcriptions with the original audio [39] and data import into Atlas.ti software. Two independent coders who were assigned to each transcription carried out an initial reading to become familiar with the discourses and start discovery of emerging topics. They compiled lists of possible topics in an open, free and creative way that laid the foundation on which to develop the successive categories, codes and memos.

Codes were generated from an inductive approach [40]. In the coding process, we distinguished two different cyclical stages: the first coding cycle, to identify and label units of meaning, and a second coding cycle when the initial codes were reorganised and reconfigured to develop a clean list of categories, topics and/or concepts [44]. The initial code tree was presented for discussion by the whole research team, agreeing on a definite list of codes, categories and subcategories as a basis for revision—and re-codification if necessary—of all interviews and field notes. Special consideration was given to non-coded fragments to identify discourse units that could be significant, and assign them the corresponding codes. We coded positive and negative “incidents” related to each code [40,44]. Possible relationships between codes and categories were noted as memos in Atlas.ti version 6.2 for subsequent consideration in interpretation of the phenomenon [42].

Finally, relativisation of the data meant their interpretation in the context and way they were collected, searching for possible influences of the interview style or the interviewer in the setting, along with interviewers’ notes and the coders’ reflexive analysis [40].

### 2.7. Trustworthiness

To ensure credibility and validity of the findings, different strategies were used during data collection and analysis to maximise the methodological rigour, according to the reliability criteria proposed by Lincoln and Guba [42,45] (See Appendix A). This study has been reported according to the Consolidated Criteria for Reporting Qualitative Research (COREQ) (See Appendix A).

## 3. Findings

Finally, members of the investigation team personally contacted 11 participants who worked at their same units. The sample characteristics are described in Table 1.

Three major themes emerged from the data: factors related to the patient, to the context and finally those related to the professionals themselves.

### 3.1. Patient-Related Determinants

The data associated with the patient’s characteristics were classified into three categories: the patient’s physical and cognitive functionality to communicate, their relational and communicative style and finally, their personal circumstances (Figure 1).

#### 3.1.1. The Patient’s Physical and Cognitive Functionality

This category comprises those factors which affect the patient’s motor and/or cognitive ability to communicate. Among the elements that exclusively influence the motor ability, most professionals identified the type of artificial airway as a limitation to verbal expression. As one nurse pointed out (P6, Table 2), the facility of lip-reading increases in patients with a tracheostomy instead of an orotracheal tube.

Another communication barrier is the lack of auditory or visual prostheses, which limits the patient’s understanding as well as his expression of messages. Furthermore, lack of dentures changes the mouth contour and hinders correct vocalisation of words, altering the patient’s capacity of expression and hindering understanding on the part of the professionals (P4a, Table 2).

The lack of physical strength stands out among factors which alter motor skills of the limbs and face (P2, Table 2). Other elements that limit the patient’s use of gestures, writing or vocalisation are the following: presence of tremor, use of physical restraints, oedema of hands or face as well as position in bed (P3, Table 2).

On the other hand, the interviewees related the alteration of cognitive communication skills to delirium and the patient’s inability to retain information (P8, Table 2).

Finally, the professionals described physical or psychological discomfort (such as pain or fear) and residual effects of drugs (sedatives and neuromuscular blocking agents) as factors that generally affect the patient’s motor and/or cognitive communication ability (P5, Table 2).

#### 3.1.2. Patient’s Relational and Communicative Style

Most professionals agreed that not all awake and intubated patients communicate the same way; some refer to the patient’s personality to explain different ways of ineffective coping detected in communication failures. They range from urgent communication expressions, to refusal to interact with the carer, or even aggressive behaviour. On the other hand, they also identified more adaptive strategies: patience and perseverance (P4b, Table 2).

#### 3.1.3. Personal Circumstances

The last category of determinants relating to the patient refers to two aspects of each patient’s very own circumstances. Many professionals identified a language barrier when interpreters are required as an added difficulty, even if it is not specific for communication with the critically ill intubated patient. Some professionals pointed out that a patient’s previous intubation experience facilitates the communication process (P10, Table 2). In a similar way, elective intubation, when the patient received information on how he will wake up in the unit, allows for much better adaptation to the situation with a lack of communication.

### 3.2. Context Determinants

This subject, organised in six categories, clusters all factors specific for the ICU setting that, according to the professionals, may affect the communication process with the awake and intubated patient (see Table 3 for related quotes, and Figure 2).

#### 3.2.1. Family Presence

Most interviewees identified family as an element with positive effects on communication, because they are more familiar with the patient and facilitate the identification of his demands (P8a, Table 3). Less frequently, relatives provide devices to facilitate communication or they act as informal interpreters in case of a language barrier (P2a, Table 3).

In contrast, one third of the interviewees stated that sometimes relatives suffer anxiety caused by the situation or unfamiliarity with the environment, which interferes with the communication process (P2b, Table 3).

#### 3.2.2. ICU Inherent Characteristics: Noise, Lighting and High Technology Care

Especially in units with an open structure, noise causes the patient confusion and does not provide an appropriate, quiet environment for communication (P3, Table 3). Poor lighting is an element which influences the capacity of using certain communication strategies (boards, writing). Finally, the use and presence of high technology equipment catches the professional’s attention, prioritising control of the physical conditions before communicating with the patient (P10, Table 3).

#### 3.2.3. Time Organisation, Workload and Continuity of Care

Most professionals agree that communication with the intubated critically ill patient needs time (P4a, Table 3). The perception of a high workload involves prioritising the time spent on monitoring activities and controlling the physical conditions, and does not consider the slow communication process.

On the other hand, high turnover of the staff that care for the same patient is a factor that limits the effectiveness of communication among professionals and patients. More than half of the interviewees felt that caring for the same patient is usually a beneficial element in the communication process, as they are more familiar with the patient’s communication style, and it allows an optimal trusting relationship for communication (P4b, Table 3).

#### 3.2.4. Availability and Features of the Communication Aids

The availability of alternative communication systems varies significantly, because their use is not formalised. Communication aids are not easily available, and they are scarcely used (P6, Table 3).

Regarding the features of communication aids, some professionals pointed out that they do not suit the patients’ needs. They stated that their contents are not appropriate and that there are accessibility and usability issues due to the patient’s incapacity to point, write or properly see the figures/letters (P8b, Table 3). Moreover, these systems are slow when producing novel messages, hence they are not used because they increase the negative emotional reactions of patients (P5a, Table 3).

#### 3.2.5. Features of the Message: Kind of Message and Output Mode

In general, most professionals felt that the more atypical the patient’s message is, the more difficulties they have to understand its meaning. Professionals have preconceptions regarding the most frequent demands, such that when the contents of the message escape those limits, the range of the patient’s potential demands broadens, thus hindering their identification (P5b, Table 3).

Regarding the output mode of the message, professionals pointed out that it is easier to understand when the message is concise, short and uses keywords (when mouthing) or capital letters to make the writing more legible (P9, Table 3).

#### 3.2.6. Communication Situations

The kind of situation where the communication process takes place is another element that impacts on the communication process. Communication happens almost always in contexts where patient care and prescribed actions take place: during hygiene or other techniques or usual procedures.

### 3.3. The Professional’s Determinants

Professionals’ determinants were classified into two categories (see Table 4 and Figure 3): professional experience on the one hand and, on the other, the person-centredness. Person-centeredness is identified as the practice approach that influences professional decisions and actions, as well as their relationships with service users and their families.

#### 3.3.1. Professional Experience

This category is tentatively emerging in the discourses of some of the most experienced professionals, who relate years of experience with higher skills to decentre attention from more physical care and extend it to other aspects such as communication.

#### 3.3.2. Person-Centredness

The professionals’ discourse identified fragments that show two opposite positions in person-centredness: the holistic approach, and the biotechnological approach, focused on technical and physiological aspects of care. Both positions are located at the ends of a continuum, in a way that some professionals vary their person-centredness very little, while others shift from one position to the other, and their discourse varies even with contradictions (pseudo-holistic approach) (see P7a, Table 4).

The influence of person-centredness on communication with the intubated patient is linked to the professional’s beliefs, which determines their communicative and relational style (Figure 3). In the following sections, both subcategories are developed.

##### The Professional’s Relational and Communicative Style

Most interviewees agreed that not all professionals communicate in the same way with awake and intubated patients. Each professional’s relational and communicative style is determined altogether by the skills, attitudes and knowledge regarding the study phenomenon.


*The professional’s skills*


Nursing professionals refer to a wide range of skills that facilitate communication with awake and intubated patients. One of them is the ability to identify the needs expressed by the patient using strategies such as the use of keywords or lip reading (P6a, Table 4).

Other skills mentioned as useful are empathy, facilitating a suitable communication environment or promoting the patient’s comprehension of the transmitted information (for example, firstly catching his attention, speaking loud and clear). Finally, some professionals state that ineffective communication and the consequent stressful reactions derived from it are fed back; hence, a capacity to break this vicious circle is highly useful (P7b, Table 4).


*The professional’s attitudes*


Among the professional attitudes that facilitate the communication process, more than half of the professionals highlight patience and perseverance as two necessary elements (P2, Table 4). Another positive attitude is acting as promoters of communication exchange, offering communication aids, facilitating communication with relatives or starting and keeping up communication with the patient even when there is no feedback (P7c, Table 4).

At the other end, a lack of prioritising in communication is one of the most influential negative attitudes in ineffective professional–patient communication (P3, Table 4). One of the participants (P8, Table 4) recognised a lack of reflection on the issue, when asked about possible solutions to communication failure.

Finally, other negative attitudes that emerged in the interviewees’ discourse were defensive position (to protect against the frustration caused by the communication failure), predisposition to failure or restriction of the patient’s communication attempts, or delaying communication until the moment when the patient can speak (P7d). Paternalist attitudes emerge when sometimes, to avoid a patient’s anxiety, the professional carries out actions that could be labelled as “communicative placebos”, transmitting to the patient the achievement of an effective communication, when this is not the case (P7e, Table 4).


*The professional’s knowledge*


The most notable fact was the paucity of comments on this theme. In the interviewees’ discourse, there were no references to the need or to the existence of protocols in the units, to the knowledge of augmentative and/or alternative communication methods (beyond boards and writing), or to related training programs.

##### The Professional’s Beliefs

The second element linked to the professional’s person-centredness is the set of beliefs regarding communication with the awake intubated patient. They are related to the communication concept, factors that influence it and their consequences.


*Beliefs related to the communication concept*


One of the most widespread beliefs in the interviewees’ discourse is the one referring to the patient’s capacity to understand the professionals’ messages. Most professionals take for granted that they express themselves clearly. As a result, *professionals perceive that communication failure depends basically on the patient’s expressive capability* (P4a, Table 4).

Other interviewees state that *caring for a sedated intubated patient is much easier than doing the same to an awake patient with limited verbal expression*. A nursing assistant (P1, Table 4) noted that many professionals prefer caring for a sedated patient to avoid stress caused by communication failure.

Another belief is that *not all messages expressed by the patient are equally significant.* Although they recognise that the patient probably perceives every transmitted message as important, this does not necessarily match the professional’s perception. In general, from the professional’s point of view “significant” messages have to do with physical stability and situations of vital risk. Other messages related to comfort, personal preferences or delusions are considered as less significant or urgent (P6b, Table 4).


*Beliefs related to factors that impact on communication*


More than half of the interviewees pointed out that *the personality of one of the conversation partners* (patient or professional) may impact on the success or failure of the communication process, the same as establishing a *good interpersonal relationship (P10 and P4b,*
Table 4*)*.


*Beliefs related to the communication result*


In the professionals’ discourse, the belief that the awake intubated patient’s needs are mainly covered despite the communication difficulties, is frequent. According to several interviewees, the demands for physical care and requests for typical information can be easily identified and satisfied without any trouble. On the other hand, there is hardly any reference related to the emotional or psychological demands.

## 4. Discussion

The findings of this study allow us to understand in depth different factors that take part in the communication process between awake, intubated patients and nurses. Some of those communication determinants were already pointed out in previous studies, especially those related to the patient’s characteristics and environment. However, this research has allowed us to deepen our understanding of communication determinants related to nursing professionals. New aspects such as the professionals’ communication style or their own beliefs have strongly emerged from the analysed data, and yield a new framework to enable to build more effective communication strategies.


*Patient-related determinants*


Among the patient-related communication determinants, those which impact on physical and/or cognitive functionality are widely reported in the literature, such as physical weakness, supine position, level of consciousness, delirium, use of physical restraints, or absence of sensorial or dental prostheses [7,12,13,21,30,31,46,47].

An innovative finding of this study is the presumed influence of physical and/or psychological comfort in the communication process, the way that the interviewees associate pain, feelings of fear or despair with a greater risk of communication failure. These results partially converge with those of Khalaila et al. [28], who correlated high levels of fear and anger in patients who cannot speak with the use of fewer communication methods. The authors did not define the directionality of this relationship, thus future studies are necessary. Nevertheless, previous literature, together with our findings, suggest a feedback system between communication failure and its emotional consequences; as patients accumulate communication failures, their frustration and stress levels rise, and this leads to abandonment of future communication attempts.

Another emerging element related to patient-linked factors is the identification of the patient’s relational and communicative style as a communication determinant. Although there are few references in the literature, Broyles et al. [46] already reported that relatives of critically ill patients based their expectations of communication on their beloved’s prior personality and communication style.


*Context-related determinants*


Many contextual communication determinants reported by the interviewees had already been identified in previous studies. This is the case for the low availability of communication aids [21,47], or their low adequacy or usability [13,48]. This could explain in part their scarce use in ICUs [8,49], especially the high-technology devices.

Regarding the content of the message, Radtke et al. [48] identified that nurses considered it more difficult to understand “novel” messages related to emotions, thoughts or decisions about the proper treatment, as they were atypical and more complex. The patients interviewed and referred by Carroll [13] insisted more in communicating when the message was important; otherwise, they tended to give up attempts to communicate.

Furthermore, in a systematic review, the influence of workload and staff shift organisation were referred to as communication constraints [47], as they are related to other pro-communicational elements such as care consistency and familiarity with the patient’s usual communication style and contents [13,14,21]. Regarding family presence, several authors highlighted its effect on communication with the intubated patient in an ambivalent sense that this promotes communication as well as restricts behaviour and/or attitudes [21,46,47].

On the other hand, there are other contextual elements that have not been previously analysed in depth, such as high-level technology, which was prominent in our findings. Discourses reflect this as a higher prioritisation of physical needs and technical care aspects as well as a communication pattern that always relates to situations involving physical care. This tendency significantly contributes to the setting aside of communication by professionals [47,48,49].


*Professionals’ determinants*


Finally, communicative constraints related to professionals form the most consistent determinant group of our results. The literature already evidences the influence of the professionals’ skills, attitudes and knowledge on communication with intubated patients [47]. In a general way, Nilsen et al. [29] detected an association between the professionals’ pro-communicative behaviour and an increase in the intubated patients’ communicative behaviour and strategies. Thus, the professionals’ training in skills and knowledge of augmentative and alternative communication is identified as a decisive factor [8,47], and is related to greater success in the communicative exchanges between professionals and patients [48,50].

Regarding professional attitudes, some authors point out that a lack of prioritising communication is a negative attitude that hinders the communication process [19,49], as our findings indicate. Moreover, the present study sets the basis for investigating other determining attitudes such as self-protection, predisposition to failure or communication restriction. Recently, Tolotti et al. [22] pointed out that nurses implement avoidance strategies when they are not able to understand what their tracheostomised patients say.

The professionals’ beliefs regarding the communication process with the intubated patient have emerged as strong determinants in this study, yielding an innovative strategy to tackle the problem. The fact that nurses prefer to care for sedated and intubated patients to avoid negative emotions derived from communication failures was already stated in previous studies [19,51]. On the other hand, other beliefs have not been confirmed in the literature, and thus constitutes a new area of constraints that requires further investigation. The interviewees’ discourses showed the existing relationship between the professionals’ beliefs and attitudes (see Figure 3). Beliefs such as thinking that, despite communication failure, the patient’s main needs are covered, that communication failure depends mainly on features that can hardly be changed, or that the patient’s messages are not always important, contribute to the idea that the communication problem is difficult to tackle, frustrating, and is placed on a secondary priority level. This lack of prioritising increases under the influence of other contextual variables such as high-level technology or elevated workloads, especially when the professional feels that communication failure requires an important investment of time and effort. In a similar manner, Handberg and Voss [49] highlighted the strong influence that staff’s cultural beliefs in a biomedical setting such as the ICU have on care values and practises regarding communication with intubated patients. As several authors stated recently, a cultural shift is necessary for a proper approach to communication with critical patients who are mechanically ventilated [52].

Communication with intubated patients seems to be strongly determined by person-centredness [53] at the same time as it is related to the professional’s skills, attitudes and beliefs. This person-centredness ranges between a more patient-centred communication at one end, and a more biotechnologically centred communication at the other, even with the same professional, as McCormack et al. [54] point out. Professional competence on one hand and the care setting on the other are determining constructs of a patient-centred care model [54]. When putting into practice this model in a critical care setting with intubated patients, communication is an indispensable tool for developing patient-centred care processes.

## 5. Limitations

One limitation of our study may be the potential transferability of the findings, circumscribed to contexts with features that are similar to those of the studied units. Finally, the perspective we offer in this research is that of the nursing professionals who care for awake intubated patients; still, there are certain convergences with other studies mentioned above where patients were interviewed. It would be interesting to complement the results with the patients’ and their relatives’ own perspectives of the situation.

## 6. Conclusions

This study reports on some aspects that have been relatively underexplored to date, such as the attitudes and beliefs. Discovery of the relationship between the different kinds of determinants (of patient, context or professionals) yields a multi-factor perspective of the communication issue. Unavoidably, this new vision must be considered when designing new approaches to improve communicative effectiveness, and suggests integrative actions aimed at changing the professionals’ beliefs.

## 7. Relevance to Clinical Practice

Little progress has been made to ameliorate communicative failure in intubated and awake patients in intensive care in the last 30 years. Most ICU nurses have not changed the way they address this problem, despite the technological advances currently available. The present research is a new step to discover the current communicative determinants in intubated and awake patients, from the perspective of the nursing professionals who take care of them.

## 8. What Does This Research Contribute to the Wider Global Clinical Community?

This study highlights the influence of the professionals’ determinants as a key element to approaching the communication problem with awake intubated patients.There is a close relationship between the professionals’ beliefs and their attitudes towards communication with those patients; it contributes to seeing it as a problem that is difficult to tackle, frustrating, and is a secondary priority.The identified multi-factor model allows the design of individualised strategies for improving the communication with these patients, overcoming the barriers identified and promoting communicative facilitators.

## Figures and Tables

**Figure 1 healthcare-11-02645-f001:**
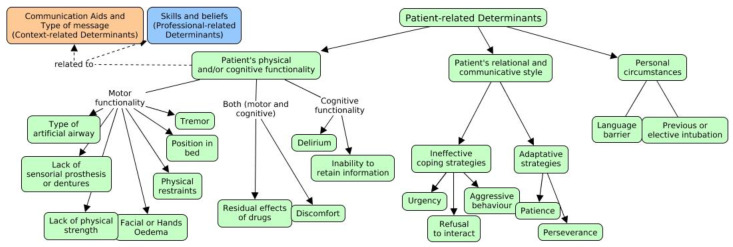
Patient-related determinants. Solid arrow: relations within theme, categories, subcategories and codes. Dot arrow: relation with other determinants, belonging to other themes (context or professional-related determinants).

**Figure 2 healthcare-11-02645-f002:**
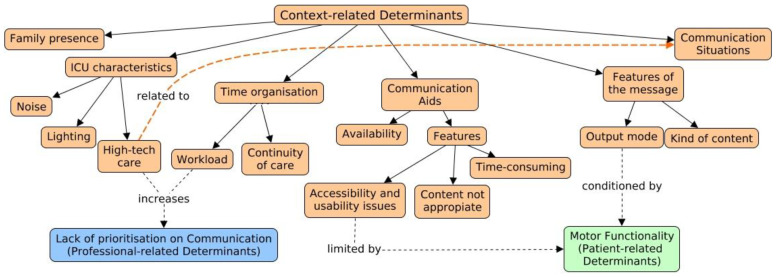
Context-related determinants. Solid arrow: relations within theme, categories, subcategories and codes. Orange Dot Arrow: relation between determinants beloging to the same theme (context-related determinants). Black Dot arrow: relation with other determinants, belonging to other themes (patient or professional-related determinants).

**Figure 3 healthcare-11-02645-f003:**
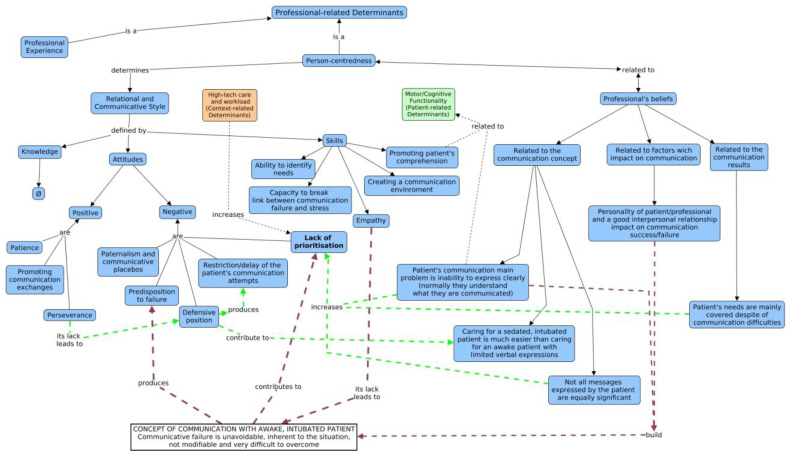
Professionals’ determinants. Solid arrow: relations within theme, categories, subcategories and codes. Green Dot Arrow: relation between determinants belonging to the same theme (professional-related determinants). Black Dot arrow: relation with other determinants, belonging to other themes (patient or context-related determinants). Purple dot arrows: Relations between the concept of communication with awake, intubated patients constructed by professionals, and other communicative determinants.

**Table 1 healthcare-11-02645-t001:** Participants’ characteristics.

Informant Code	Profession	Age	Gender	Professional Expertise (Total Years)	ICU Expertise (Total Years)	Years at Present ICU	Hospital	Coders
P1	Nursing Assistant	27	Male	9	8	7	Hospital 1	PC/MM
P2	Nurse	27	Female	6.5	5	5	Hospital 1	PC/AT
P3	Nurse	39	Male	16	14	9	Hospital 1	PC/GG
P4	Nurse	33	Female	10	6	6	Hospital 2	PC/GG
P5	Nurse	42	Male	10	10	10	Hospital 2	PC/GT
P6	Nurse	35	Male	7	5	2	Hospital 3	PC/MM
P7	Nurse	49	Female	27	26	26	Hospital 3	PC/MM
P8	Nursing Assistant	28	Female	9.5	7	7	Hospital 2	PC/GG
P9	Nursing Assistant	42	Male	12	7	7	Hospital 3	PC/AT
P10	Nursing Assistant	37	Female	12	11	11	Hospital 3	PC/AT
P11	Nursing Assistant	31	Female	13	10	6	Hospital 1	PC/GT

**Table 2 healthcare-11-02645-t002:** Quotes related to patient’s determinants.

Patient’s Physical and Cognitive Functionality
Tracheostomy vs. orotracheal tube
“because [with tracheostomy] the lip mobility is complete and […] articulation is much better” (P6).
Lack of sensory aids/dentures
“many times […] they are elderly people, they don’t have teeth, you know? […] their cheeks are […] inwards and you don’t understand them [when mouthing]” (P4a).
Lack of physical strength
“Patients who don’t have strength, who are bedridden, they can’t use their hands to write or to point on the communication chart nor make any gestures” (P2).
Other factors limiting motor function
“[…], of course, an oedematous patient, badly seated, unable to write well, who […] then asks for pen and paper and writes […] trying to use good handwriting, but he can’t do it well, this makes communication still more difficult” (P3).
Alteration of cognition
“when they are intubated, you don’t know whether they understand much of what you explain them; you’re trying to tell them something and they say “yes”, but after a while they’re back with the same matter and you realise that they haven’t understood it; so you don’t know whether they’re confused or not” (P8).
Discomfort
“State of mind, an anxious or restless state, pain […] has much to do, sometimes [the patient’s] anxiety makes communication [with him] impossible” (P5).
Patient’s relational and communicative style
“I think that the […] inpatient’s personality […] has a great influence. There are very anxious people, so then, when you try to advise them that they can try to communicate with you in some form, there is no [no way] […]. You clearly notice when a person is calm or has got a smooth character, then there arrives a moment when his eyes tell you “well, you don’t understand me, don’t bother, it doesn’t matter” (P4b).
Personal circumstances
“for example, most [COPD patients] who had been admitted many many times […] know the process very well, some even talk with the tube in place, which means that you understand them quite well” (P10).

**Table 3 healthcare-11-02645-t003:** Quotes related to context determinants.

Family Presence
“sometimes I couldn’t understand, didn’t know what the patient says, and then his family immediately say “see, it’s about this or that” […]. Of course, they know him well” (P8a). “Many times it’s the family who brings in a whiteboard” (P2a). “Yes, […] when the patient tries to speak, the ventilator beeps and […] [relatives] become more anxious, so often they tell the patient “don’t speak, don’t speak, don’t say anything” (P2b).
ICU inherent characteristics
“at sometimes there forms such a noise that it makes [communication] difficult […] as there is no peace to be able to listen to [or understand] him” (P3). “Also, ICU is a place where they use so much technology, they use many techniques and most professionals don’t see the patient […] as a person, they see [him] as a disease that requires certain technologies and they administer them, that’s all” (P10).
Time organisation, workload and continuity of care
“sometimes, when the workload is high, you have so much to do, you’re there trying to understand what he wants to ask, […] you don’t understand, you’ve got things to do […] and you don’t have time” (P4a). “the longer they have stayed in, the easier it becomes to understand them, that’s true” (P4b).
Availability and features of the communication aids
“[regarding the communication board] I don’t even know whether there is one in every unit and it must be you who looks for it in every drawer until you finally find it. There is no clearly assigned place, not everybody uses it, far from using it every day […] when you ask for it, you don’t find it” (P6). “In this alphabet there are a few pictures, but far too little. Maybe there is some fruit or a pen... that’s very little. […] Patients hardly ever go to the pictures, maybe because it is a small alphabet with small pictures; I’m quite sure that more than half [the patients] don’t see properly. […] Perhaps if we had bigger charts [...] it would be easier than with those small ones” (P8b). “they want to write the whole sentence on the board and as it works letter by letter they become more desperate” (P5a).
Features of the message
“It’s because you go with preconceived ideas […] and ask him whether he feels comfortable, if he’s in pain, hungry, if […] he’s cold, these are the questions you’re going to ask the patient; […] [if] the patient answers that he wants to see his son, […] how can he make you understand that he wishes to see his wife or his family, when you don’t have the same preconception?” (P5b). “please write it in capital letter so we can see it” (P9).

**Table 4 healthcare-11-02645-t004:** Quotes related to professional’s determinants.

Person-Centredness (Pseudo-Holistic Approach)
“[during hygiene] I like to ask them whether it bothers them when we [the professionals] talk about our matters, because […] it seems as if you didn’t pay them any attention, you see? And I think that it’s important to ask them, but we don’t always do it, you see? But, mainly when they are more awake, “do you mind if we talk about this or that?” […] “does it bother you if we talk about that?” “No”, because he, well, he watches and is aware of it too” (P7a).
Professional’s skills
“[The patients] try to communicate and you begin […] “What’s the matter? Pain?” […] “What troubles you? The tube?” […] we have more experience in lip reading and we feel what problem or trouble they may have […] I make sure to have all those factors under control so it’s not them that cause the need of communication, and then I have to draw on him writing or me trying to read his lips or ask questions” (P6a). “When you see that he becomes nervous, you stop; […] my method is to wait for a while and then I start again… [...] you wait a while until he calms down; [...] I think that [it’s appropriate] to stop a little and then start again” (P7b).
Professional’s attitudes
“other [professionals] […] who spend more time, have more patience, stay longer by the bed until they achieve to understand them” (P2). “You always start with “good morning, good afternoon, how did you spend the afternoon, do you remember me?”, you see? In a way, starting to talk almost as if he didn’t have the tube, you know? And you don’t wait for his answer, but you keep on talking. […] When the family is there, the family doesn’t understand him and then they become nervous because maybe he asks something of the family, something personal, from outside the hospital, you see? Then you must help them a bit and mediate also between family and patient. It’s a little like this […]” (P7c). “In general, standing beside the patient in order to talk with him is not really a habit we have [the professionals]” (P3). “Well I don’t know in which way [it could be improved], the truth is that I never have thought about it” (P8). “don’t worry, within two days they will remove you the tube, don’t worry” (P7d). Many times… they want to write, and writing doesn’t always yield positive results […] but, well, to calm them down, you let them write, but most of the times […] you don’t understand what they write, you know? Or many times you see some scribbling, […] but, why, sometimes they calm down because […] they wrote their discourse, you know? […] don’t ever tell them that they [cannot write], because they need to try, and then you have to tell them that yes, they did it, they wrote, and then they calm down: “yes, I understand”, “ah, OK”, you only need to say “ah, OK”, even if they haven’t written anything, you see? and they calm down a little” (P7e).
Professional’s beliefs.
“to me it’s much more difficult to understand them than to explain myself, because I can tell them everything I’m doing to them, why I’m doing… ” (P4a). “it’s more complicated because… they should be sedated and they aren’t, and of course they ask many questions you don’t understand and you don’t know how to answer” (P1). “many times [patients] are confused and you think that they want to tell you something important, but it’s a result of their disorientation” (P6b). “I think that the patient’s personality and that of the professionals’ involved with the patient have an impact, as communication is not the same with one professional or another” (P10). “Let’s see, sometimes the professional’s character also clashes with the patient’s character, […] there are very patient nurses and others with less patience” (P4b).

## Data Availability

The data presented in this study are available upon request from the corresponding author. The data is not publicly available because the transcribed interviews are protected by Organic Law 3/2018, of 5 December, on the Protection of Personal Data and guarantee of digital rights.

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
