# Peer review of "Determinants of Communication Failure in Intubated Critically Ill Patients: A Qualitative Phenomenological Study from the Perspective of Critical Care Nurses"

_healthcare, 2023, doi:10.3390/healthcare11192645_

Round 1
Reviewer 1 Report
thankyou for the opportunity to review your manuscript.
introduction.
mostly well structured to provide context to the study. The last paragraph that discusses the aim needs to be clearer. "What factors determine communication of awake intubated critically ill patients..." is not clear in terms of what your study is trying to do - what are you tryin gto determine? effective communication? ineffective communication? failure of communication? determine communication difficulties?
methods.
sample size was originally set to 8 and final number was 11. This is not clear in the methods. Why did you stop at 11? Was there thematic saturation? Were there issues with recruitment?
what is the difference between a nursing assistant and a nurse?
the rest of the methods are well written.
findings.
the number of quotes makes it difficult to read the content. having this as a supplemntary with some common themed quotes would be better.
discussion.
a well written and balanced discussion of findings. It's refreshing to see a discussion that argues both sides of the argument and uses other evidence to prove. well done.
conclusion.
broad, in keeping with findings.
grammatical errors throughout eg the first sentence of the conclusion "...some aspects little investigated up to date".
Author Response
Dear reviewer,
We sincerely appreciate your valuable revision of our manuscript. Your feedback is extremely helpful to us.
We will now address your comments point by point and make the necessary changes in the manuscript document (highlighted in red). Thank you once again for your time and effort in reviewing our work.
Please see the attachment.
Best regards,

Reviewer 2 Report
Dear authors,
I have revised your manuscript. I think that some major points should be addressed.
· Introduction (you can merge introduction and background): One notable weakness in this introduction is the absence of a clear and cohesive structure. It frequently introduces new ideas and shifts between discussing the problem, its effects on patients, and the rationale for conducting research in a somewhat disjointed manner. Each paragraph often appears to be based on a single phrase or concept, resulting in an incomplete exploration of important aspects of the topic. This lack of a well-defined structure can make it challenging for readers to follow the logical flow of ideas and may hinder overall readability. To enhance the introduction, a more organized and systematic arrangement of information is needed, allowing for a smoother transition between different aspects of the research topic. This would aid in maintaining reader engagement and facilitating a better understanding of the research context.
· While it mentions using a phenomenological approach, it lacks a strong justification for choosing this specific methodology. A clear rationale for why phenomenology is the most suitable approach for addressing the research question should be provided (in the subheading regarding design). In relation to the participant recruitment, the section briefly mentions participant recruitment through purposive sampling but does not elaborate on how this process was conducted in practice. Providing more details about the sampling process, such as how participants were identified and selected, would enhance transparency. Likely, the description of data collection is somewhat vague. It mentions using semi-structured interviews but does not provide information about the interview questions or how the interviews were conducted beyond their average duration and how the guide for the interviews was developed (e.g., “previous elaboration of the initial interview script was based on a review of the existing literature” but the review was not cited). Additionally, details about how data saturation was determined are lacking.
· Findings: What you have called “factors related to the patient” are the patient determinants? It is not clear, please be consistent in using terms. Overall, this section is clear.
· Discussion: this section discusses the findings of the study but lacks clarity in presenting them. It contains multiple points without clear organization or structure. To enhance readability, the findings should be presented in a more organized and structured manner, possibly using subheadings or bullet points to highlight key findings.
· Overall: check the reference style that should be in accordance with the journal’s guidelines.
Author Response
Dear reviewer,
We sincerely appreciate your valuable revision of our manuscript. Your feedback is extremely helpful to us.
We will now address your comments point by point and make the necessary changes in the manuscript document (highlighted in blue). Thank you once again for your time and effort in reviewing our work.
Best regards,

Round 2
Reviewer 2 Report
Well done